# Roles of Histone H2B, H3 and H4 Variants in Cancer Development and Prognosis

**DOI:** 10.3390/ijms25179699

**Published:** 2024-09-07

**Authors:** Po Man Lai, Xiaoxiang Gong, Kui Ming Chan

**Affiliations:** Department of Biomedical Sciences, City University of Hong Kong, Hong Kong SAR, China; pomanlai2-c@my.cityu.edu.hk (P.M.L.); xiaoxgong3-c@my.cityu.edu.hk (X.G.)

**Keywords:** epigenetics, chromatin, histone variants, variants, cancers, H2B, histone 2B variants, H3, histone H3 variants, H4, histone H4 variants

## Abstract

Histone variants are the paralogs of core histones (H2A, H2B, H3 and H4). They are stably expressed throughout the cell cycle in a replication-independent fashion and are capable of replacing canonical counterparts under different fundamental biological processes. Variants have been shown to take part in multiple processes, including DNA damage repair, transcriptional regulation and X chromosome inactivation, with some of them even specializing in lineage-specific roles like spermatogenesis. Several reports have recently identified some unprecedented variants from different histone families and exploited their prognostic value in distinct types of cancer. Among the four classes of canonical histones, the H2A family has the greatest number of variants known to date, followed by H2B, H3 and H4. In our prior review, we focused on summarizing all 19 mammalian histone H2A variants. Here in this review, we aim to complete the full summary of the roles of mammalian histone variants from the remaining histone H2B, H3, and H4 families, along with an overview of their roles in cancer biology and their prognostic value in a clinical context.

## 1. Introduction

The eukaryotic genome is highly compacted into chromatin in a well-organized manner due to the presence of enormous numbers of repeating nucleosomes [1]. As the fundamental subunit of chromatin, each nucleosome entails a 147 bp (base pair) long double helix DNA that wraps in nearly 1.7 turns around a histone octamer and is formed upon the assembly of two H2A-H2B dimers and an H3-H4 tetramer [2,3]. In addition, in higher eukaryotes, linker histones H1 and H5, which bind to the DNA in between the nucleosomal complexes, are crucial proteins that assist in stabilizing the compaction of nucleosome arrays into a higher-order chromatin structure [4]. H1 and H5 work by bringing the two linker DNA strands close together to form a unique stem-like structure, therefore further facilitating chromatin compaction [5,6,7].

During DNA replication, in addition to genomic DNA, it is also necessary for core histones to be duplicated and quickly assembled on newly synthesized DNA daughter strands to consolidate the unstable chromatin structure. To achieve this, two different processes that are tightly coupled to DNA replication are involved in the chromatin construction process [8]. The first reaction is the direct transfer of histone proteins from parental nucleosomes onto either one of the two DNA daughter strands without preferences or the synthesis of new replication-coupled histones, followed by their assembly on the daughter strands with the assistance from a multitude histone modifying enzymes and histone chaperones, with CAFs (Chromatin Assembly Factors) histone chaperone being the classic and most studied one [9,10,11]. These replication-dependent core histones are encoded by three gene clusters located separately on chromosome 6 (6p21-6p22), chromosome 1 (1q21) and chromosome 1 (1q42) [12,13]. Despite their role in maintaining the stability of chromatin, several modifying mechanisms such as post-translational modification of histones, ATP-dependent nucleosome remodeling and substitution of histone variants with specific chaperones for core histones can cause dynamic changes to the chromatin structure and, in turn, regulate gene expression [14].

Histone variants are paralogs that can replace core histones to form nucleosomal complexes. They were found to be constitutively expressed throughout the cell cycle, while their exchanges within chromatin are aided by specific remodelers and histone chaperones to guarantee an accurate nucleosome deposition. Variants’ protein sequences differ from canonical histones to a large degree, ranging from a few amino acids to large non-histone domains. Similar to core histones, variants can undergo PTMs (Post-Translational Modifications), and thus, their replacement can alter the nucleosomal stability [15,16]. As a result, the substitution of variants can support diverse fundamental biological processes such as DNA damage repair, chromosomal segregation, dosage compensation and transcriptional regulation. Additionally, some variants have acquired a lineage-specific task, like spermatogenesis, throughout evolutionary development [16,17,18,19].

In the past few decades, a promising number of reports have laid the foundation for the elucidation of histone variant functions and have revealed that histone variants, despite originating from different histone families, perform overlapping tasks in addition to their major duties. Recently, emerging reports have demonstrated that histone variants are potential driving causes of various cancers in which they take part at different cancer progression stages, including aggressive cell proliferation, migration and metastasis. In addition, growing evidence has suggested that histone variants may also act as novel biomarkers and/or prognostic indicators for a wide range of cancer types and early detection. However, since the roles of some newly identified or uncovered histone variants remain obscure, further research is required to investigate and explore their potential roles in cancer progression.

As the histone H2A family has the largest number of mammalian histone variants reported to date, our prior review solely focused on summarizing each of the roles of H2A variants in fundamental cellular mechanisms, cancer progressions and their potential use in clinical settings [20]. In this review, we continue to provide a comprehensive overview of the functions of all the remaining mammalian histone variants in H2B, H3 and H4 families documented as of now (Table 1), along with their roles in cancer biology and clinical settings.

## 2. Histone H2B Variants

Histone H2B is one of the four typical histones involved in chromatin formation. Since novel variants were recently discovered by Raman et al., the number of H2B variants that can be found in mammals has increased to 14 (Table 1). Surprisingly, studies have displayed that the evolutionary rates of various H2B variants vary considerably. This implies that some H2B variants may have evolved faster, leading to notable sequential differences from canonical H2B [21]. So far, however, almost all of the H2B variants have been found to lack a strong correlation with any of the known cancer-inducing mechanisms, with only a few studies briefly touching on this topic. Alternatively, nearly all H2B variants are identified as testis-specific by primarily playing functional roles in spermatogenesis and early fertilization. Evidence also supported H2B variants’ functions in spermatozoa, as the vast majority of them have an effect on non-conventional chromatin packing or non-chromatin functions in mammalian germ cell lines [22].

### 2.1. H2B.E

H2B.E is found to regulate gene transcription and longevity in neurons and is different from canonical H2B by only five amino acid residues. H2B.E was first identified in rodents and is expressed in their olfactory sensory neurons [23]. The expression levels of H2B.E in mature olfactory neurons show a direct impact on neurons’ lifespan. As a result, inactive neurons usually express higher levels of H2B.E, and the aberrant upregulation of H2B.E is suggested to promote neuronal cell death and result in shorter lifespan [21,23]. According to its specific role in neuronal status, a recent study published in 2024 by Feierman et al. deciphered that H2B.E is broadly expressed in the brain and preferentially enriched at promoter regions, thus playing a role in promoting synaptic gene expression and long-term memory [24]. Therefore, H2B.E’s role leads to the hypothesis that H2B variants may also be involved in the pathogenesis of neurological disorders, such as Alzheimer’s disease, but more research is needed to support this viewpoint.

Intriguingly, a paper published in 2023 by Xu et al. discovered a novel role of H2B.E in viral replication. They demonstrated that H2B.E binds to Nsp9 (Nonstructural Protein 9) of PEDV (Porcine Epidemic Diarrhea Virus), and Nsp9 subsequently upregulates H2B.E expression, hence significantly enhancing viral replication through the perturbation of ER (Endoplasmic Reticulum) stress-mediated apoptosis of infected host cells [25].

### 2.2. H2B.A

H2B.A was once captured by the application of an electron capture dissociation experiment by Siuti et al. They have identified that H2B.A variant is the most abundantly expressed variant, and its expression level remains constant at different cell cycle stages [26]. In 2005, H2B.A was found to be acetylated at its lysine residues at positions 12th, 15th and 20th by Bonenfant et al. group [27]. However, it is still unclear how the PTM of this histone variant takes part or which gene loci it is enriched at to facilitate downstream events. Interestingly, H2B.A and many other proteins were discovered in the mouse pancreatic islets by Ahmed et al.’s group [28], while the altered expression of the other proteins has been proven to be strongly correlated with diabetes development, H2B.A may potentially be one of the key players in the pathogenesis of diabetes. This undoubtedly paves the way for future studies to comprehend the relationship between H2B.A and diabetic individuals and provides novel insights or first-draft references towards diabetes treatment in clinical settings.

### 2.3. H2B.W

H2B.W is found in the sperm nuclei and can be incorporated into telomeric regions [29]. H2B.W is in the same clade as H2B.M, as both of them have an extended C-terminal tail, while the major difference between the two lies in the N-terminal tail [21]. Even though H2B.W shares only less than 50% identity with canonical H2B, of note, most of the functional domains, including H2A and H4 interacting residues, are conserved. It implies that these conserved residues and modification sites may play important roles in nucleoprotein interactions and chromatin organization.

Unlike H2B, H2B.W is not able to assemble into mitotic chromosomes or recruit chromosomal condensation components. H2B.W-containing nucleosomes are easily remodeled and mobilized by the actions of the SWI/SNF (Switch defective/Sucrose non-fermentable) complex, thus increasing chromatin accessibility for rapid gene transcriptional activities [29].

There are studies showing an association between SNP (Single Nucleotide Polymorphisms) in the H2B.W encoding gene and defects in male fertility, implying that the presence of SNPs may cause reduced or defective male fertility in individuals [30]. The C to T change on the 9th base of the 5′ untranslated region of the H2B.W gene may lead to sperm rudimentary development, reduced sperm count and viability [30,31,32].

### 2.4. H2B.1

One of the earliest H2B variants discovered in mammalian testis is H2B.1 (TSH2B/TH2B) [33]. It has an 85% identity with canonical H2B and differs in 19 amino acids. The majority of H2B.1 is abundantly expressed during spermatogenesis and in fertilized zygotes, while they were also detected in mouse oocytes, with a very low amount in mouse ESC (Embryonic Stem Cell) [21,34]. However, the role H2B.1 plays in oocytes and ESC remains uncharacterized. H2B.1-containing nucleosomes are less stable, suggesting their role in facilitating histone-to-protamine packing during spermatogenesis [35]. This further implies that H2B.1 may be involved in pronuclei formation and the activation of paternal genes throughout the early stages of embryonic development and post-fertilization [36].

Apart from being present in sperm, H2B.1 experiences composition changes by decaying during postnatal development in rat cerebral cortex neurons [37]. According to this, researchers also tried to connect the H2B.1 variant with a variety of human neurological disorders or diseases. In one of the studies published by Luna et al., they observed that H2B.1 was downregulated together with multiple proteins in pediatric brain tumors, but little attention has been paid to the underlying molecular mechanisms of such an H2B.1 reduction [38]. This, again, creates a direction for future research on whether epigenetic reprogramming may contribute to cancer development or a novel therapeutic target to treat pediatric brain tumors.

In the establishment of a malignant Friend tumor, chromatin reorganization by the changes in histone variants’ composition is investigated. H2B.1 is examined to show an increased level with a concomitant decrease of H2B.2 [39]. It is also surprising to know that the lowest H2B.2 to H2B.1 ratio contributes to the most malignant cell type of Friend erythroleukemia [40]. These perspectives provide strong evidence that the involvement of histone variants in chromatin reorganization signified the progression of malignant cell progression. Further investigation is therefore crucial to shed some light on the correlation between H2B variants and cancer development and offer more inspiring insights and ideas to solve the unrevealed questions in the field.

### 2.5. Rare H2B Variants

H2B variants are not commonly studied as they have been shown to demonstrate very little relation to major diseases or cancer progression. Still, a couple of H2B variants are rarely mentioned or newly identified by some studies, with a very short and brief description. They are H2B.A, H2B.B, H2B.F, H2B.J, H2B.K, H2B.L, H2B.M, H2B.N, H2B.O, H2B.Q and H2B.2.

H2B.O is an undescribed group of H2B variants in the past decades. H2B.O has only been found to be present, albeit at a low enrichment, in platypus’ reproductive tissues, testes or ovaries [21]. We are now unable to draw more informative conclusions about its functions because of its clear absence from placental mammals. Besides, H2B.B, H2B.F and H2B.Q are only mentioned in a study done by Siuti et al., without any description of their roles in any cell types [26]. As they have not yet been the focus of new studies in the field, developing a deeper understanding of these three obscure variants that exist in our human body is essential.

H2B.K and H2B.N are two phylogenetically distinct clades that are highly distinguishable from other H2B variants, as they are encoded by intron-containing genes [21]. In most mammals, H2B.K and H2B.N are present in a single copy. However, interestingly, they were pseudogenized in rodents [21]. In addition, the introns of these two variants are located at the same position, suggesting that these variants may share a common ancestor. Also, unlike other H2B variants that are primarily expressed in male testes or sperm, H2B.K and H2B.N are instead overwhelmingly expressed in oocytes and early fertilized eggs [21]. Moreover, as H2B.K has only a few differences compared to canonical H2B, it is likely that there will be conserved functions between the two. H2B.N is dramatically different from canonical H2B with less than 50% identity, with the most significant difference being their truncated C-terminus [21]. Nevertheless, very little is known about these two recently uncovered H2B variants, which could open the door to more in-depth research.

H2B.M, also named H2B.W.2, is found within the H2B.W clade across mammalian species. H2B.M is different from H2B.W mainly due to the divergence in its N-terminal tails [21]. Raman et al.’s study is the sole report done on H2B.M with only brief information on how it differs in sequence from H2B.W. H2B.J was once revealed in Siuti et al.’s study, and intriguingly, was later discovered to possibly possess the ability to control viral infections in conjunction with other proteins. In Zhang et al.’s study, they identified that the H2B.J and viral 3C protease were targeted by PARP9-DTX3L (Poly(ADP-Ribose) Polymerase Family Member 9/Deltex E3 Ubiquitin Ligase 3L), subsequently disrupting viral particle assembly and enhancing host defense through upregulation of interferon signaling [41].

One study published in 1984 by Grove et al. discovered a dynamic expression of two histone variants (H2A.1 and H2B.2) in differentiating MELs (Murine Erythroleukemia Cells). They showed that when MELs were terminally differentiated, H2B.2 and H2A.1 rapidly increased in tandem with the number of hemoglobin-producing cells [42]. Additionally, H2B.2 is found to accumulate in rat cerebral cortex neurons during postnatal development [37]. However, since more details have not been provided, its functional role in MEL differentiation, differentiating neurons, or other cell types remains uncertain and demands further elucidation.

The last rare H2B variant identified is H2B.L, also known as subH2B/subacrosomal H2B. It is originally isolated from the sperm of bulls and rodents with abundant expression [21]. H2B.L is highly different from canonical H2B in terms of its H4 interacting residues, the L2 residue, and the post-translational modification site [21]. These differences may affect the mode of interaction of H2B.L with other proteins (e.g., H4), leading to differences in chromatin assembly and structural regulatory functions. H2B.L can be found in the sperm across mammalian species except in humans, with a preference for localization in subacrosomes in spermatozoa, but its function has not been determined [43]. While H2B.L appears to be a pseudogene in the human genome with a single frameshift mutation, this mutation is also present in gorillas, chimpanzees and many other primates [21]. This leads to the idea that H2B.L’s functions may be taken over by other preserved histone variants during the fast evolution of H2B variants.

## 3. Histone H3 Variants

Histone H3 variants are a special group of variants, the majority of which share almost 97% of the identity with canonical H3. There are a total of 11 mammalian H3 variants being documented and studied to date (Table 1). Interestingly, H3 has both replication-independent (RI) and two (H3.1 and H3.2) replication-coupled (RC) variants [22]. RC variants are tightly coupled to DNA replication and only deposited or exchanged within the chromatin during the S-phase, which is similar to what canonical H3 does. The main differences between RI and RC H3 variants are found in the 87th to 90th amino acid residues (SAVM/AAIG in mammals) of the N-terminal end of alpha-helix 2. Therefore, they are regulated by diverse specific histone chaperones and are able to cause different modes of chromatin assembly [44]. For instance, H3.1 and H3.2 are regulated by CAF-1 (Chromatin Assembly Complex 1) and RBBP4 (RB Binding Protein 4), whereas H3.3 is regulated by HIRA (Histone cell cycle regulator A), ATRX (Alpha Thalassemia Mental Retardation Syndrome X-linked) and DAXX (Death Domain-Associated Protein) [44,45,46]. Considering their structural disparities, RC and RI H3 variants indeed play unique and irreplaceable roles in a wide range of fundamental cellular processes in mammals.

### 3.1. H3.1 and H3.2

H3.1 and H3.2 are mammal-specific RC paralogs of canonical histone H3 [47]. These two variants only differ in one amino acid in mammalian species and were first identified in 1977 [48]. It was later found that these two variants’ incorporation into chromatin is highly coupled with DNA replication, and therefore, they were named canonical/replication-dependent histone variants [49]. Despite the two canonical histone variants conserving opposing functions, many reports have demonstrated the functions and roles of H3.1 in various contexts, including cancer. On the other hand, sporadic studies were performed on H3.2 and were hence discussed most of the time together with H3.1 in the majority of published studies.

H3.1 and H3.2 are found to be especially enriched and mark the inactive and late replicating regions in mammalian cells [50,51]. H3.1 variants contain an additional cysteine 96 in their sequences; therefore, H3.1-nucleosomes can bind to other nucleosomes containing H3.1 through an inter-nucleosomal disulfide bond between the two cysteines, hence assisting a higher-order nucleosomal structure of the heterochromatin [52]. H3.1 is commonly acetylated or demethylated on lysine 9 and 14, whereas H3.2 undergoes a slightly different N-terminal modification with a high preference for H3.2K27me3 (lysine 27 trimethylation) [53]. H3.1 and H3.2’s deposition is regulated by a specific histone chaperone—CAF-1, which is a nuclear complex comprised of three subunits: CHAF1A (p150), CHAF1B (p60) and p48 in humans [54,55].

In a study reported by Tagami et al., H3.1 complexes in pre-deposited form within mammalian tissues were purified along with the additional identification of H4, as a common component, together with HAT1 (Histone acetyltransferase 1), CAF-1 and two closely related histone chaperones—ASF1A (Anti-silencing Factor 1A) and ASF1B in H3.1 complexes [46,56]. These proteins synergize with CAF-1 to perform nucleosome assembling by a mechanism linked with DNA synthesis during DNA repair in human cells. Interestingly, H3.1-H4 complexes are deposited first on DNA as a dimeric unit rather than a hetero-tetramer. Therefore, later studies concluded that the formation of tetramers was indeed a two-step mechanism [57]. It is suggested that the remaining H3 complexes are either (1) completed by the CAF-1 subunit p150 that exhibits dimerization properties [58] or (2) assisted by the presence of two chaperones, CAF-1 and ASF1, in the free H3.1 complexes to accomplish nucleosome assembly behind the replication fork during DNA replication and DNA damage repair [59]. However, this hypothesis requires further elucidation in future studies.

Although most of the literature published thus far has focused mostly on the functional role of H3.1 in plants, the amino acid sequence of H3.1 in mammals only differs from plants by one residue. Consequently, scientists suggested that the mechanisms involving H3.1 similar to those found in flowering plants, such as regulating DNA repair, were possibly conserved in mammals. TSK/TONSL (Tonsoku-like, DNA repair protein), a protein required for HR (Homologous Recombination) repairment during DNA damage, is vital for maintaining genome stability by minimizing DSBs (Double Strand Breaks) throughout our genome during stressed replication forks and other forms of DNA damage in human cells [60,61]. The conserved TPR (Tetratricopeptide Repeat) domain, with extensive similarity in plants’ TSK orthologs, is located in the N-terminal of mammalian TONSL and exhibits a stronger preference for interactions with H3.1 variants. Therefore, these domains are subsequently recruited onto the chromatin through its ARD (Ankyrin Repeat Domain) to resolve any broken replication forks throughout DNA replication [62,63]. Both plants and mammals also share similar mechanisms for post-replicative chromatin maturation. In mammals, the binding of TONSL via the ARD domain is interfered with by SET-8 (SET domain-containing protein 8)-mediated mono-methylation of the 20th lysine residues on H4 histone [62,64]. This can ensure that HR-mediated DNA repair does not occur outside the S and G2 phases of the cell cycle.

Distinct epigenetic modifications at the 87th, 89th and 90th residues of H3.1 are essential for specifying different chromatin localization and nucleosome assembly pathways of H3.1 and other H3 variants in mammalian cells [65,66,67,68]. H3.1–H4 complexes are found to be more preferentially acetylated by HAT1 [65]. When HAT1 is depleted, H3.1 complexes’ association with CAF-1 and importin 4 is reduced, leading to alterations in the expression of certain H3.1-enriched genes that subsequently affect the regular cell cycle progression [46,65]. Besides, ATXR5 (Arabidopsis Trithorax-Related Protein 5), a H3 lysine 27 methyltransferase, can specifically bind to H3.1 and regulate K27 methylation, which in turn provides a platform for mitotic inheritance and genome-wide distribution of repressive marks in both plants and mammals [69,70].

In addition, H3.1 and H3.2 also participate in early embryonic development and replication in mammals. H3.1/3.2 were found to be deposited in both euchromatin and heterochromatin regions in mESCs (Mouse Embryonic Stem Cells) [53]. One of the studies done by Kawamura et al. identified an asymmetrical incorporation of H3.1/3.2 into male and female peri-nucleolar regions of one-cell embryos [71]. This incorporation is important as the DNA replication timing will otherwise be delayed and cause developmental failure if H3.1/3.2 are forcefully incorporated into the paternal pro-nuclei of the one-cell embryo. Furthermore, Gatto et al. showed that H3.1/3.2 and H3.3 boundaries were linked to the maintenance of stability regarding the early replication zones in human cells during the S-phase [72].

Finally, H3.1 also plays a role in regulating meiotic heterochromatin condensation through DNA polymerase epsilon. This is observed in both plants and mice, suggesting a highly conserved mechanism across different kinds of eukaryotes. Polymerase epsilon has a catalytic subunit POL2A, which possesses two different domains: (1) a C-terminal zinc finger domain (ZF1) that is responsible for binding to H3.1–H4 complexes residing in heterochromatin regions and (2) an N-terminal that specifically recruits and guides MORC1 (MORC Family CW-Type Zinc Finger 1) to its localization site for inducing a continuous meiotic heterochromatin condensation process [73].

Altogether, H3.1/3.2 cooperate with a wide range of proteins and play crucial roles in regulating multiple biological processes, including nucleosome assembly, mitotic inheritance and meiotic heterochromatin condensation in mammalian cells, which ultimately help maintain and safeguard genome stability.

#### 3.1.1. Biological Significance of H3.1

A study conducted by Meszar et al. pointed out an interestingly pivotal role of the H3.1 variant in assisting the maintenance of mammalian thermal sensitivity toward burn injury [74]. They utilized mouse models with the application of the CRISPR-Cas system and viral transduction of mutated H3.1 in which S10 (Serine 10) phosphorylation was blocked. It was discovered that the responses of excitatory dynorphinergic (Pdyn) neurons to burn-injury tissue damage heavily depended on the H3.1S10 phosphorylation-dependent signaling, as when mutated H3.1 was transduced into the mice, their acute thermosensation ability was severely lost, with a significant increase in thermal nociceptive threshold [75].

#### 3.1.2. Biological Significance of H3.2

One study carried out by Vetrivel et al. illustrated an ocular regulatory function specific to the H3.2 variant. The *Hist2H3C1* gene, encoding H3.2 variants, was found to be mutated and highly enriched in a small-eye mutant Aey69 mouse model. The elevated expression of H3.2 also affected the Ephrin signaling pathway. They later clarified H3.2’s functional significance as a crucial epigenetic player that binds to particular gene regions to support healthy early ocular development [76]. The functional role of H3.2 in normal eye development was also interpreted by another recent study, showing that the 358th amino acid Ile to Leu (A to C) mutation of the H3.2 encoding gene in mice could lead to lens vesicle degeneration and severe microphthalmia [77].

#### 3.1.3. H3.1 and H3.2’s Roles in Cancers—Oncohistone Variants

Multiple reports have documented H3.1 mutations and their role in various types of cancer, with a main focus on brain cancers. Lysine 27 to methionine (K27M) mutation is a characteristic of pediatric DMGs (Diffuse Midline Gliomas). H3.1K27M only accounts for 20% of K27M mutations, as most of these are found in H3.3 [78,79,80,81]. Besides, H3.1K27M has also been observed in DIPGs (Diffuse Intrinsic Pontine Gliomas) and non-brain stem gliomas [82]. H3.1K27M inhibits genome-wide H3K27 trimethylation only in cells undergoing cell cycle progression through perturbing and poisoning PRC2 (Polycomb Repressive Complex 2) complexes [82,83,84]. Therefore, the disruption of PRC2 complexes’ functions may lead to an alteration in the epigenetic profiles and gene expression regulation, eventually predisposing cells to tumorigenesis (Figure 1).

In other cancers like AML (Acute Myeloid Leukemia), H3.1 mutations, including H3.1K27M and H3.1K27I, have been identified [83]. These mutations strongly correlate with the decrease in H3K27me3 and are commonly associated with *RUNX1* (Runt-related Transcription Factor 1) gene aberrations. It has been suggested that these mutations are one of the accelerators of AML progression by being predominantly involved in the loss of p53 activity and activation of the RTK (Receptor Tyrosine Kinase) signaling pathway (Figure 1) [85].

Recent studies have found that the H3.1K27M mutation was also associated with distinct oligodendroglial cell lineages, including oligodendrocytes, ependymal cells and astrocytes [86,87]. H3.1 mutations originating from different lineages can directly affect the intra-tumoral transcriptional activities and tumor morphology. A subset of BMGs (Brain Midline Gliomas) bearing H3.1K27M mutations specifically appeared to be an ependymal-like population of cells. These cancer cells uniquely express genes and proteins that belong to the ependymal transcriptional programs, for example, FOXJ1 (Forkhead Box Protein J 1) transcription factors [88,89]. This type of H3.1K27M BMGs may also preferentially associate with the activation of PI3K (phosphatidylinositol-3-kinase) pathways and somatic ACVR1 (Activin A Receptor Type 1) mutation. This gives cells an oncogenic property and promotes tumorigenesis through constitutively activating the BMP (Bone Morphogenetic Protein) signaling pathway, arresting glial differentiation and inducing mesenchymal phenotypes by downstream activation of Stat3 (Signal Transducer and Activator of Transcription 3) (Figure 1) [88,90,91,92].

In addition, other mutations such as H3.1E97K, which can only be minimally incorporated into the chromatin, can still alter the nucleosomal stability and enhance the colony formation ability of cells to promote cancer progression [93]. Studies have shown that H3.1E97K-H4 complexes simultaneously dissociated through examination under thermal stability assay, pointing out that the mutant H3.1-containing nucleosome was unstable and might even compete with wild-type H3.1, causing perturbations in the proper functions of H3.1 regarding the maintenance of genome stability [93].

Besides H3.1 mutation, polyadenylation of H3.1 mRNA induced by exposure to environmental carcinogens such as arsenic, nickel and bisphenol can lead to an abnormal reduction of SLBP (Stem-Loop-Binding Protein) by proteasomal degradation and transcription inhibition [94]. This consequentially leads to an increase in polyadenylated H3.1. However, excessive expression may sensitize the cells to additional DNA damage, causing metal or bisphenol-induced toxicity and carcinogenicity [95,96,97]. Interestingly, the underlying mechanism for arsenic-mediated polyA tail formation on H3.1 variants remains ambiguous, and more insights into this mechanism should be provided.

H3.1 is also involved in the reprogramming of cells during metastatic transition. When a cancer cell line is induced to show EMT (Epithelial Mesenchymal Transition) and metastasis, H3.1 mRNA and protein levels are significantly downregulated, whereas chromatin accessibility is increased [98]. The decrease in H3.1 allows the incorporation of H3.3 variants mostly into the promoter of EMT, cancer progression and metastasis-related genes [49]. The H3 variants’ dynamic, therefore, heavily fuels metastatic progression in different cancer cells. A few studies have pointed out that H3.1 may also indirectly promote metastasis. *NUP210* (nucleoporin 210), a susceptible gene with its metastasis potential confirmed in Amin et al.’s study, potentially interacts with H3.1, which is commonly seen in patients with poor prognosis in breast cancer. The interactions between the two may exert an effect by prohibiting the heterochromatin modifying enzymes from reaching locations where the H3.1 variant resides, hence leading to the opening up or dysregulation of silenced genes in metastatic cell lines [99]. Additionally, a recent study revealed that the H3.1 variant is also a chromatin redox sensor and can indirectly aid breast tumor cells in gaining an advantage in both chemoresistance and metastatic properties [100].

On the other hand, H3.2 mutation is rarely documented, with only a few studies exploring its oncogenic role in *Drosophila* rather than mammalian species. H3.2K36R mutants were found to be able to silence Polycomb activities, perturbing H3K27me3 processes, which in turn interfere with the expression of cell type-specific genes. Moreover, combinatorial H3.2 and H3.3 variant mutations can guide *HOX* gene misexpression and lead to impaired early larvae development [101]. However, due to highly conserved H3.2 sequences among different eukaryotes throughout evolution, it is somehow believed that H3.2 mutant may also induce a similar effect on mammalian species by causing defects in H3K27me3 spreading through Polycomb silencing [101,102].

In addition, H3.2’s oncogenic role was described in a study reported by Reddy et al. Transcript levels of H3.2 variants were aberrantly elevated, with concomitant downregulation of H3.3 variants in HCC (Hepatocellular carcinoma), whereas H3.1 levels remained unaltered. Interestingly, heightened H3.2 expression was observed across different cancer cell lines that were tested, and this was also increased in gastric cancer, as reported by Rashid et al. [103]. H3.2 was suggested to enhance repressive marks in tumor cells, especially where tumor suppressor genes with the potential of governing cell proliferation resided [104]. More studies should be conducted on H3.2 variants in the future to provide a better understanding of its underlying mechanisms in various cancer cell lines, as well as its prognostic value and potency of becoming a biomarker or indicator in different cancer types.

#### 3.1.4. Prognostic Significance of H3.1

Given that H3.1 mutations are regularly reported and examined in many research contexts and are typically associated with a poor prognostic outcome in cancer patients, H3.1 can thus offer some prognostic value. Based on what was previously stated, the majority of the H3.1 mutations detected were linked to brain tumors, such as DMGs. Pediatric patients frequently experience H3.1 mutations, while children who have more H3.1 mutations than H3.3 mutations unexpectedly tend to have longer survival. Oppositely, adult patients with more H3.1 mutations result in poor outcomes and shorter survival rates [105,106]. Therefore, it is suggested that regular H3.1 mutation genotyping is essential for diagnosis and may act as an indicator for different age groups [107]. In addition to that, combined genotyping of both H3.1 and H3.3 mutations can potentially double the efficacy of diagnostic processes. Surprisingly, a recent study published by Grassl et al. revealed a groundbreaking H3K27M-targeted, first-in-human vaccine for DMG patients with a maximized therapeutic benefit [108]. More detailed clarification is crucial for the combinatorial immunotherapy and cancer field, which may raise both the effectiveness and efficacy of cancer treatments.

Besides genotyping the H3.1 variants’ mutations, liquid biopsy has received more attention in recent decades due to its minimally invasive capability. A few studies suggested detecting H3.1K27M in urine, cerebrospinal fluid and blood as one of the biomarkers and indicators for cancer detection and monitoring [109]. As H3.1K27M is commonly found in pediatric DMGs, measuring the aberrant epigenetic profiles of ctDNA (circulating tumor DNA) may provide better insights into the prognostic value of cancer stages and progression. Apart from DMGs, other reports also suggested that H3.1 variants can act as a biomarker for hematological malignancy—non-Hodgkin Lymphoma [110]. However, H3.1K27M mutation is quite common in most cancer types, thus solely measuring the epigenetic profiles of cell-free circulating nucleosomes may not yield the highest specificity and sensitivity. Therefore, future research is required to solve this unmet need.

### 3.2. H3.3—Multifunctional Histone Variant

H3.3 is a replication-independent histone variant and is encoded by two different genes, *H3F3A* and *H3F3B,* in the human genome [111]. They differ from canonical histone H3 by only four amino acid residues at the 31st, 87th, 89th and 90th positions. The differences in amino acids at S87A, V89I and M90G are mainly responsible for the variation between H3 and H3.3’s integration into chromatin [44]. H3.3 is specifically recognized by three different histone chaperones, including HIRA, ATRX and DAXX [45,46]. Interestingly, the functional roles of H3.3 are highly dependent on the type of histone chaperones. HIRA-mediated distribution of H3.3 is mostly found to incorporate promoter regions with high GC content, transcriptional-activated genes and enhancer regions for histone turnover during transcription [112,113]. While H3.3 nucleosomal complexes are less stable than H3 nucleosomes, as seen in different studies, they can facilitate the binding of subsequent transcription factors to enhance targeted gene expression [114]. Following the deposition of ATRX and DAXX-mediated H3.3 distribution primarily into telomeres, imprinted genes and even heterochromatin regions, H3.3 is then modified with lysine 9 trimethylation marks [115,116]. This strongly suggests that H3.3 enables transcriptional enhancement, heterochromatin maintenance and telomere structure remodeling.

Besides regulating gene activities, H3.3 also plays a critical role in fertilization, mouse embryogenesis, cell differentiation and pluripotency. A few studies have discovered a dynamic switch pattern between canonical H3 and H3.3 during oogenesis and two cell stages [117]. In addition, H3.3 turnover rates are globally elevated near the TSS (Transcription Start Site) regions and are found to be much higher in differentiated cells compared to ESC (embryonic stem cells) [118]. This further proves that H3.3 participates in gene regulation, including developmental-specific genes, to maintain the delicate balance between gene activation and repression at different stages. In terms of fertilization, the protamines wrapped around sperm chromatin are gradually replaced by H3.3 variants residing on female chromatin in the oocyte, resulting in male chromatin de-condensation and contributing to chromatin reprogramming [119]. These results demonstrate that the timely transition of H3.3 in different stages is significant in supporting the developmental program of early embryos [120].

Furthermore, H3.3 also assists in normal brain development. Consistent with its role in cell differentiation, H3.3 is an important epigenetic regulator for NSC (Neuronal Stem Cells) proliferation and differentiation [121]. With the suppression of H3.3, premature termination of neuronal cell mitosis and PAX6 (Paired box protein Pax-6) positive NSCs differentiation are obviously observed [121]. In addition to that, H3.3 directly interacts with a specific H4K16 acetyltransferase, MOF, in which the acetylation modification of NSCs is simultaneously reduced after the knockdown of H3.3. Moreover, GLI1 (Glioma-associated oncogene), a transcriptional regulator, is also concomitantly downregulated in H3.3 knockdown NSCs [121]. Taking into account the aforementioned properties, H3.3 cooperates with other proteins in precisely regulated embryonic brain development. In addition to its role in brain development, H3.3 is also found to maintain adult HSCs (Hematopoietic stem cells) homeostatic hematopoiesis by preventing cell death [122]. Also, H3.3 is responsible for balancing the survival, cell renewal and different lineage differentiation of HSCs through the delicate interplay between different epigenetic modifications.

#### 3.2.1. H3.3’s Role in Cancers

Multiple studies have illustrated the mutation of the H3.3 variant and its corresponding functional role in tumorigenesis. Similar to H3.1 and H3.2, most of the H3.3 mutations are found to be mostly correlated with different subtypes of brain tumors, followed by giant cell tumors in bone, while the dysregulated modification of H3.3 can also contribute to HCC (Hepatocellular carcinoma) and lung cancers.

Different single amino acid substitution mutations at diverse positions were investigated and examined in DMGs (Diffuse Midline Glioma), pHGGs (Pediatric High-grade Glioma), chondroblastoma and other types of gliomas. In pHGGs, most children and young adults are found to harbor H3.3G34R/V mutations [123]. These mutations, interestingly, are age-restricted whilst specific to pHGGs, and are considered as the driving cause of pHGGs’ malignancy and therapeutic resistance [124]. The underlying mechanisms of these mutations are thought to be the reprogramming of epigenetic markers on the regulatory element genes associated with different downstream signaling pathways, including JAK/STAT (Janus Kinase/Signal Transducer and Activator of Transcription), DNA repair pathways and cGAS/STING (Cyclic GMP–AMP Synthase/Stimulator of Interferon Genes) pathways [123,125,126]. The hyperactivation of these pathways can lead to an alteration in the tumor immune microenvironment and an overall dysregulated intrinsic environment. In addition, G34R mutation results in enhanced susceptibility towards DNA damage and the inhibition of the cell’s DDR (DNA Damage Response), subsequently leading to heightened accumulation of mutations and induction of immune cytokines storm (Figure 2) [123]. In current therapeutic interventions, maximal treatment efficacy in pHGGs patients is achieved by administering combinatorial treatments, including radiotherapy, downstream targeted inhibitors/agonists and immune-mediated gene therapy. These can effectively boost their immunological memory, successfully prolong one’s lifespan and survival time and, more importantly, prevent relapse of pHGGs [123,125].

In addition to H3.3G34R/V mutations, the H3.3K27M mutation is identified in DIPGs (Diffuse Intrinsic Pontine Gliomas) and DMGs. K27M mutation results in a widespread reduction of lysine 27 trimethylation of histone H3 proteins and eventually drives gliomagenesis by disrupting normal PRC2-mediated gene silencing patterns [82,127,128,129]. Interestingly, H3K4me3 modification was observed to increase in the promoter region of some specific tumor-related genes, such as the neural transcription factor OLIG2. The upregulation of OLIG2 could inhibit the p53 signaling pathway and promote malignant gliomagenesis (Figure 1) [130]. This suggests that the local epigenetic changes caused by the K27M mutation facilitate the abnormal expression of tumor-related genes and provide a growth advantage for tumor cells.

Several recent studies have determined that H3.3K27M mutation can also disrupt the PML (Promyelocytic Leukemia) nuclear body formation, leading to a stalled differentiation, apoptosis and continuation of stem-cell-like cell proliferation phenotype, which is found to be similar to that in APL (Acute promyelocytic Leukemia) [131]. Besides, both H3.3K27M and H3.3G34R/V can enhance chromatin accessibility and cause aberrant oncogenes upregulation associated with neurogenesis and NOTCH signaling pathway, thereby sustaining tumor progression (Figure 1 and Figure 2) [132,133].

Not only limited to brain tumors, the H3.3G34W mutation is found to be specific to a vast majority of the GCTBs (Giant Cell Tumors of Bone), a type of bone cancer, while minor subsets of GCTBs harbor H3.3–G34L/M/R/V mutations [134]. How the G34W mutation causes bone cancer was explored in multiple studies. Forsyth et al.’s report indicated that the integration of mutated variant into the chromatin causes serious genome instability, worsened telomere maintenance through hypomethylation of heterochromatic regions and reactivation of hTERT together with the shortening of telomere’s ends [135]. Moreover, H3.3G34W affects both intrinsic and extrinsic cell biological processes. Macrophages with upregulated RANKL (Receptor Activator of Nuclear Factor Kappa-B ligand) in their cell surface, in GCTB’s tumor microenvironment, secrete MCSF (Macrophage Colony Stimulation Factor) and display an extensive combination of receptors with low OPG secretion, which in turn produce a large number of large osteoclastic giant cells responsible for osteoclastogenesis in GCTB (Figure 3) [134,135].

Another histone H3 mutation, the H3.3K36M missense mutation, is mostly identified in chondroblastoma [136]. Studies exploited the consequences of this mutation and observed a collapse in normal K27 trimethylation along with a widespread reduction in K36 demethylation [137]. In fact, these altered modifications misregulate gene expression, causing defects in chondrocyte differentiation and enhanced colony formation ability of mutant cells, leading to subsequent tumorigenesis induction [138]. mHGAs (Midline High-Grade Astrocytomas) are another type of pediatric cancer caused by a H3.3K27M substitution mutation in the downstream signaling receptor, ACVR1 (Activin A Receptor Type 1), which causes subsequent hyperactivation of BMP (Bone Morphogenetic Protein)–ACVR1 developmental signaling pathways, and ultimately upregulates downstream early-response genes in tumor cells (Figure 1) [92].

Undeniably, mutation is the most common and direct reason for carcinogenesis; however, changes in gene expression levels also play a significant role in disease progression and development. Multiple studies discovered a high correlation between aberrant overexpression of the H3.3 variant and lung cancer progression, confirming that such overexpression is essential for the acquisition of cancer cells’ migration and metastasis. The molecular mechanism behind this is that H3.3 preferentially occupies a specific intronic region of GPR87 (G Protein-Coupled Receptor 87) and directly regulates its expression [139]. Of note is that GPR87 has shown its potential in driving metastasis and mediating immunogenomic landscape in lung cancer, as reported in the past decades [140]. Moreover, upregulation of the H3.3 variant was found to promote ARMS (Alveolar Rhabdomyosarcoma) metastatic traits, with an effect on enhancing cell invasion and increasing Rho activation [141]. Growing evidence has also suggested that the dynamic incorporation of the H3.3 variant is commonly found in metastasis-inducing transcription factors or genes and results in poor prognosis in invasive carcinomas, which is consistent with what was mentioned previously [98,142]. Alternatively, the downregulation of H3.3 and synchronous upregulation of H3.2 are observed in HCC. The dynamic change between histone variants favors the gene repression of various tumor-suppressive genes, thus possibly enhancing uncontrollable liver cancer cell proliferation and contributing to HCC development [104].

#### 3.2.2. Prognostic Significance of H3.3

As H3.3 mutation is found to be associated with many aggressive carcinomas, this positions it as a potential and robust biomarker or indicator for cancer diagnosis. Accordingly, a number of studies concluded from their IHC (immunohistochemistry) staining that different H3.3 mutations contribute to a distinctive morphology of cancer cells. For instance, the H3.3K36M mutation is highly specific to chondroblastoma, and IHC using a chondro-related antibody can effectively distinguish real chondroblastoma from other types of cancer, including suspected chondroblastoma, GCTBs or cartilage matrix-diffuse bone tumors [136]. This effectively raises detection efficacy and sensitivity. Similarly, the H3.3G34W mutation is a common and highly unique mutation found in GCTBs, and a 77.8% sensitivity in patients with GCTBs can be achieved by just measuring the expression of the H3.3G34W mutation [143]. The specificity of H3.3K36M in chondroblastoma and H3.3G34W in GCTBs have been further proven by other studies using fine-needle aspiration and core needle biopsy [144].

Of note, in terms of DMGs, the prognostic value of H3.3K27M is largely influenced by factors such as patient age. Pediatric cancers with more H3.3 mutations are usually associated with poorer survival rates [107]. Conversely, adults who bear more H3.3 mutation tend to have prolonged survival rates and better prognostic outcomes. As previously mentioned, the H3.3 mutation and the H3.1 mutation exist in an incompatible relationship. It is therefore suggested that both mutational rates should be detected to examine DMGs’ progression status, and this can be a prognostic indicator for patients’ survival rate. More importantly, H3.3K27M is absent in low-grade gliomas, embryonal tumors, or other extracerebral pediatric solid tumors, which can extensively raise its specificity and sensitivity. However, the identification of H3.3G34R/V mutations in pHGGs, DMGs and other CNS–PNETs (Central Nervous System—Primitive Neuroectodermal Tumors) are relatively rare, so G34R/V mutations may not be suitable or capable of serving as biomarkers in cancer detection [145].

Besides, multiple studies have revealed dysregulated expression levels of H3.3 in different cancers, such as HCC, lung cancers and ARMS. Inspired by different research on the detection of circulating tumor DNA (ctDNA), this minimally invasive tactic can be applied to detect H3.3 from patients’ blood samples [146]. It is feasible to infer that patients may be highly prone to suffer from undetected cancer if a greater number of H3.3 variants are found, but further research is necessitated to fully understand and consider its application value in early cancer detection.

Due to the location and difficulties in infiltrative detection of brain tumors, the urgent need for safe and non-invasive detection methods remains unmet in the field. More advanced research is required to improve the current diagnostic approaches for patients, ideally with minimal intrusion.

### 3.3. H3.4—Development of Early-Stage Spermatogenesis

H3.4 is a mammal- and testis-specific H3 variant. It can be found in rats and mice and has several alternative names, including H3t and H3.1t [22,147]. This variant gene is located on chromosome 1q42.13 in humans and on chromosome 11 in mice [19,148]. Surprisingly, it has been shown that H3.4 shares a common chaperon recognition motif (SAVM) at the 87th to 90th amino acid sequences with both H3.1 and H3.2 and is therefore, also a replication-dependent variant. H3.4 is solely expressed and transcribed in the testis, with its functional significance being investigated by Ueda et al.’s group through the generation of H3t knockout mice using the CRISPR/Cas9 gene editing technique. The male mice with H3t-/- genetic background eventually became infertile [19]. Additionally, null H3.4 mutation remarkably results in reduced testis size, and therefore, transparent or no spermatozoa were found in the epididymides.

H3.4 is specifically induced only when the differentiating spermatogonia emerge, as H3.4 is completely absent in mature spermatozoa [149]. This suggests that H3.4 is highly likely to be replaced by other H3 variants, mainly H3.3 and/or protamines, during spermatogenesis [150,151]. In addition, H3.4-containing nucleosomes form an easily opened-up chromatin structure compared to the canonical H3 nucleosomes, substantially providing a platform for germ cells to enter spermatogenesis. Fascinatingly, upon examination of H3.4-deficient germline stem cells, H3.4 is deciphered to assist the germline stem cells in terms of self-renewal, differentiation and meiosis entry properties [19,152]. Until now, studies on H3.4 have only been limited to spermatogenesis development, according to the published papers. Thus, whether H3.4 is involved in any type of malignancy is still unknown and requires further interpretation.

### 3.4. H3.5—Testis-Specific Histone Variant

H3.5, a histone H3 variant predominantly expressed in human seminiferous tubules of testes, was discovered in the past decades. H3.5 is suggested to have evolved from a common ancestor, the great apes, and is encoded in the human chromosome 12p11.21 [153]. Documented literature related to H3.5 variants is scarce, with reports mainly focusing on their role during sperm development. H3.5 is only found in immature germ cells and is ascertained to preferentially accumulate at the transcription start site as well as other euchromatin regions of actively transcribed genes in testicular cells during normal spermatogenesis [154]. Interestingly, comprehensive studies reported by several groups have illustrated that ectopically expressed H3.5 is incorporated into chromatin to form an unstable nucleosome [154]. The destabilization of the nucleosome is due to the presence of a specific Leucine residue at position 103 in H3.5 that weakens the hydrophobic interaction with histone H4 [154]. Thus, it was learned that incorporation of H3.5 may induce an open chromatin confirmation during early spermatogenesis to allow the expression of stage-dependent genes for normal sperm maturation. As H3.5 plays a critical role in early spermatogenesis, this suggests that H3.5 might be regulated by gonadotrophin, a male hormone that acts as an essential regulating factor and may thus serve as a therapeutic target for spermatogenic disorders or other gonadotrophin-regulated diseases [155].

Surprisingly, H3.5 has an unexpected ability to replace and compensate for the functions of H3.3-deficient cells in cell growth [153,154]. The overlapping function between H3.5 and H3.3 is rarely examined and remains obscure. To better understand the implications of variant dynamics in carcinogenesis, early cancer prognosis or detection, and concomitant treatments, more insights should be given to the event of H3.3 deficiency. Whether or not H3.5 will take over the role of H3.3 and contribute to malignancy progression serves as a future subject for elucidation.

Intriguingly, as ectopically expressed H3.5 is frequently exchanged into the chromatin and causes an unstable nucleosome, a few studies have emphasized on examining the role of the H3.5 variant in tumor cells. Reports from Kandoth et al. and Urahama et al. revealed several missense mutations, including Val-100 and Arg-130 on *H3F3C*, the H3.5 encoding gene, as well as inappropriate expression of H3.5 mRNA levels in cancer cells [154,156]. This will thus be an inspiring foundation for understanding the correlation between aberrant production of H3.5 and cancer progression.

### 3.5. H3.6, H3.7 and H3.8—Human-Specific Histone Variants

Three novel human tissue-specific histone H3 variants have been newly identified. To date, not many studies have described their functional roles and applications in different scenarios. H3.6, H3.7 and H3.8 have been annotated as pseudogenes for decades, and recently, Taguchi et al. provided insights on these three histone H3 variants by examining their presence in multiple human tissues, including the ovary, heart, thyroid, and prostate and more [157]. Additionally, they discovered that the H3.6 nucleosome is unstable, in which the nucleosome destabilization character is solely caused by the Val62 amino acid residue of H3.6. While H3.8-containing nucleosomes have low thermal stability, H3.7 has failed to form any nucleosomes in vitro [22,157,158]. H3.6, H3.7 and H3.8 are encoded by *H3F3AP6*, *HIST2H3PS2* and *H3F3AP5* genes respectively. H3.6 and H3.8 are considered H3.3 derivatives due to a few amino acid differences from H3.3 encoding sequences, whereas H3.7 is regarded as an H3.1 derivative with four amino acid residue differences.

To date, only one report by Taguchi et al. has described H3.6—our understanding of H3.6 is, therefore, limited to its ability to destabilize the nucleosomal complex due to Val62 residues. Future studies are required to expand our knowledge of H3.6 to comprehend its significance in the eukaryotic genome and its altered expression levels in different corresponding human tissues may contribute to different diseases, including cancers.

A few studies have described H3.7’s functional role in *Stylonychia*. However, none of the published articles have described its applications in mammalian species. Our understanding of H3.7’s role is restricted to its involvement in the development of macronuclei during sexual reproduction in *Stylonychia*. Studies have mentioned that H3.7 expression is highly enriched in macronuclear anlagen and is often acetylated [159]. This modification is observed to be prevalent in a class of sequences that are exclusively kept in mature macronuclei, indicating that it may play a role in regulating the fate of specific sequences to aid in genome processing during macronuclear development [160].

Apart from H3.7, H3.8 expression was also discovered in *Stylonychia* and is alternatively found to be the only H3 variant detected in micronuclei. H3.8 commonly undergoes lysine methylation and threonine/serine phosphorylation [159]. However, the functional role of H3.8 in micronuclei has not been investigated in detail, as the only study done on human H3.8 was limited to the confirmation of an unstable nucleosomal complex by cryo-EM and biochemical analyses [158].

While there is a dearth of reports introducing these three H3 variants, the discoveries of their distributions in various human tissues may establish a new topic for future elucidation of their importance, as well as of the consequences of their mutational or post-translational modifications, will provide fresh perspectives on the pathogenesis or etiology of certain diseases and disorders.

### 3.6. H3.X and H3.Y—Novel Primate-Specific H3 Variants

H3.X (H3.Y.2) and H3.Y (H3.Y.1) are unique to Hominidae and are located on human chromosome 5p15.1 [147]. H3.X and H3.Y are highly similar to each other and differ by only four amino acid residues. The 87th to 90th amino acids are an important region for chaperone-mediated chromatin incorporation of variants. Since this region is identical across H3.3, H3.X and H3.Y, it is therefore assumed that HIRA also mediates H3.X and H3.Y chromatin incorporation [22,147,161]. The expression of H3.X and H3.Y are induced by a transcription factor, DUX4 (Double Homeobox Protein 4), which is expressed during an early cleavage stage embryo for controlling the zygotic gene transcriptions [162]. Consequently, H3.X and H3.Y are incorporated into DUX4-mediated genes, potentially regulating the expression of genes related to zygotic development.

These two histone variants are detected in various parts of the brain, including the hippocampus, cerebellum and cerebral cortex. As H3.X and H3.Y proteins are only expressed in a subpopulation of neurons, this highlights that H3.X and H3.Y may also possess cell type-specific functions, but this requires further investigation. Interestingly, Wiedemann et al. observed the presence of H3.X and H3.Y mRNA transcript levels in various human malignant tissues, such as human osteosarcoma, lung, ovaries, and breast tumor tissues. However, only H3.Y proteins were detected in vivo [163]. As the reason for the absence of H3.X proteins remains obscure, H3.X is hypothesized to (1) act as a pseudogene with no functions, (2) bear unknown regulatory mechanisms, (3) in some way impacting H3.Y expression, or (4) be stage-dependently expressed during zygotic development.

Additionally, by further exposing starvation and overgrowth stress stimuli to U2OS cell lines, more H3.X and H3.Y mRNA are expressed, with more H3.Y being present [163]. Consistently, only endogenous H3.Y proteins are found in the stress-treated U2OS cells. Previously, a study explored the preferential residence of H3.Y nucleosome at TSS. Based on this, H3.Y was knocked down in U2OS cells, which resulted in a significant reduction in cell growth. This highly suggested that H3.Y was responsible for gene expressions related to cell cycle control and cell growth [147,163,164]. As the stress stimulation partially mimics the tumor growth environment with rapid cell proliferation and competition for nutrients; moreover, different tumor samples showed higher H3.Y levels; these results strongly proposed that H3.Y acts as an oncogenic variant that assists tumor cell growth by modulating cell cycle control. However, the validity of this suggestion demands supporting evidence and further research. Collectively, these observations intriguingly pave the way for further investigation on H3.Y’s functional roles in malignancies and may also lay the groundwork for its usage as a new diagnostic biomarker/indicator or as a target for cancer treatment. It would also be interesting to study the connection between H3.X and H3.Y, as H3.Y depletion opposingly increases H3.X mRNA.

### 3.7. CENP-A—Variant-Specific to the Centromeric Regions

CENP-A (Centromere Protein A) is the only histone H3 variant found to replace canonical H3 in the functional centromeric region. CENP-A-containing nucleosomes wrap only 121 base pairs of DNA around their cores, making them more compact than other H3 variant nucleosomes [165]. This plays a crucial role in the maintenance of genome integrity by recruiting centromere-specific proteins to form a complete kinetochore complex, thus enabling the formation of its normal structure and accurate chromosome segregation during cell division [166]. It is interesting to know that the expression levels and localization of CENP-A are highly dependent on cell cycle regulation, and this guarantees precise timing and positioning during mitosis. At S-phase, CENP-A is bound to the centromere and is evenly partitioned between sister centromeres while maintaining a stable association across multiple rounds of cell divisions [167].

An important protein chaperone in the regulation of CENP-A is HJURP (Holliday Junction Recognition Protein). HJURP interacts with CENP-A and facilitates its deposition in centromere chromatin [168]. In addition to HJURP, CENP-I, a kinetochore subunit, also assists in the localization and stabilization of CENP-A nucleosomes during the cell cycle [169]. On the other hand, CENP-A collaborates with CENP-B in shaping the centromeric chromatin state [170]. Surprisingly, there has been little focus on the normal functions of CENP-A in mitosis and cell division during the acquisition of previous foundational knowledge. The functional role of CENP-A in cancer progression was mainly explored, as many of the malignancies are found to have an overexpression of CENP-A, followed by the alteration of downstream signaling pathways.

#### 3.7.1. Role of CENP-A in Cancers

CENP-A’s oncogenic roles have been depicted in a wide variety of cancers, including lung cancer, breast cancer, prostate cancer, renal cancer, colon cancer, HCC and many more malignancies. Overexpression of CENP-A is a feature in many cancers [171]. Altered levels of CENP-A can affect the formation of CENP-A-containing nucleosomes and centromere function, leading to mitotic defects and chromosomal instability through triggering abnormal changes, such as chromosomal deletions and translocations, which in turn exacerbate genetic instability in cancer cells [172]. With the abnormality in chromatin organization, the regulation of genes is perturbed, resulting in remodeled gene expression patterns and activation/de-activation of signaling pathways.

In lung adenocarcinoma, CENP-A has been validated as the regulator for lung tumor stem cells’ stemness by Yu et al.’s group [173]. They showed that CENP-A was involved in cell proliferation regulation, as knocking down CENP-A reduced the number of cancer stem cells. To further validate the role of CENP-A in vivo, they subcutaneously injected the transfected cells with CENP-A knockdown into immunodeficient mice and observed a decrease in tumor volume and size. In addition, CENP-A, together with CDK1 (Cyclin-dependent kinase 1) and CDC20 (Cell division cycle protein 20), are found to be highly co-expressed in lung cancer tissues [174]. CDK1 and CDC20 have been discovered to play a critical role in cell cycle regulation and spindle checkpoint pathways (Figure 4) [175]. This indicates an accelerated cell proliferation rate that eventually leads to uncontrollable cell division.

In HCC, upregulated CENP-A also increases cell proliferation and tumor growth. Furthermore, CENP-A functions as a transcriptional regulator activated following the lactylation of its lysine 124 residues to enhance target gene expression [176]. It also cooperates with YY1, which drives the expression of CCND1 (cyclin D1) and NRP2 (Neuropilin 2) to promote HCC progression [176]. Given that both YY1 and CCND1 take part in cell cycle regulation, NRP2 was found to be crucial for macrophage maturation and metastasis induction in multiple cancers (Figure 4) [177,178,179]. The results showed that the enhanced target gene expression might be correlated with the host immune response, tumor microenvironment and metastasis-inducing factors, for instance, EMT (Epithelial–Mesenchymal Transition). Furthermore, CENP-A facilitates STMN1 (Stathmin 1) transcription by binding to its promoter while suppressing the ferroptosis of HCC cancer cells, allowing rapid growth of HCC and establishing a malignant phenotype [180].

HCC can be developed due to distinct factors, including unhealthy lifestyle (high-fat diet and alcoholic diet) and HBV (Hepatitis B Virus) infection. One study published by Liu et al. in 2012 revealed that frequently mutated HBx (Hepatitis B Virus X Protein) can greatly boost CENP-A expression in HCC tissues. Nevertheless, the mechanism behind this is unknown and requires further elucidation [181].

It has been reported that CENP-A also plays a role in breast cancer progression. As mentioned above, CENP-A acts as a transcriptional regulator and promotes DNMT1 (DNA Methyltransferase 1)-mediated PLA2R1 (Phospholipase A2 Receptor) promoter methylation, thereby reducing the expression level of PLA2R1 and augmenting breast cancer progression [182]. It has been demonstrated that upon knockdown of CENP-A and overexpression of PLA2R1, breast cancer cell proliferation and migration ability are restrained with enhanced apoptosis, whereas the tumor growth and volume were also effectively suppressed upon CENP-A knockdown in vivo [182]. Additionally, Zhang et al. determined that breast cancer patients with higher CENP-A levels are more likely to experience PI3K/Akt/mTOR (Phosphoinositide 3 kinase/Protein kinase B/mammalian Target of Rapamycin) intracellular signaling pathway activation, boosting cancer cell growth and chemotherapy resistance (Figure 4) [183,184].

In TNBC (Triple-Negative Breast Cancer), one of the breast cancer subtypes, CENP-A, is thought to play a pivotal role together with a transcription factor, FOXM1 (Factor Forkhead Box M1), and other glycolysis-related genes in TNBC proliferation, cell migration, metastasis and metabolism reprogramming (Figure 4) [185]. These phenotypes, therefore, contribute to heightened aggressiveness and a higher risk of distant metastasis in TNBC.

In renal cell carcinoma, CENP-A’s oncogenic roles have been depicted in both PRCC (Papillary Renal Cell Carcinoma) and ccRCC (clear cell Renal Cell Carcinoma). In PRCC, CENP-A expression levels are significantly correlated with the pathological tumor stages and progression. Thus, patients with higher CENP-A expression often have a worse prognosis. From the functional analysis, CENP-A was discovered to be enriched in pathways related to extracellular matrix regulators and glycoproteins, neuroactive ligand receptors and cytokine-receptor interactions [186]. Other signaling pathways are alternatively activated by CENP-A in ccRCC. Wang et al. stated that the Wnt/β-catenin pathway was upregulated by overexpression of CENP-A, consequently accelerating ccRCC proliferation and metastasis through the upregulation of cell cycle division [187]. Similarly, in both prostate cancer and epithelial ovarian cancer, increased CENP-A levels also influence cancer development through the ING cell cycle (Figure 4) [188,189,190,191].

CENP-A was found to impact some immune pathways, such as cytokine-receptor interactions. Yang et al. also determined that aberrant expression of CENP-A in gliomas surges immune cell infiltration. Furthermore, the inflammation-related interactions (Interleukin 6/Janus kinase/Signal Transducer and Activator of Transcription protein) signaling pathway is hyperactivated, turns out to perturbing anti-tumor immune responses to encourage tumor progression (Figure 4) [192].

For endometrial cancer, ectopically expressed CENP-A upregulates SLC38A1 (Solute Carrier Family 38 member 1), leading to the enhancement in metabolism reprogramming and glutamine uptake, which subsequently contribute to its aggressiveness and poor prognosis [193]. CENP-A also strengthens glycolysis in colon cancer by acting as an upstream transcription activator of an oncogene, *KPNA2* (Karyopherin α2 subunit), that is known to be involved in metabolic reprogramming in cancer. CENP-A recruits histone acetyltransferase, GCN-5, to *KPNA2*’s promoter regions to induce transcriptional activation, thereby augmenting colon cancer development (Figure 4) [194]. Surprisingly, there are distinct classes of CENP-A hotspots, including 8q24, that exist in the sub-telomeric chromosomal locations, and the accumulation of CENP-A at 8q24 can be seen in the early stage of primary colorectal tumors [195].

The subnuclear localization of CENP-A is significantly and abnormally altered in tumor cells compared to normal cells. In normal cells, CENP-A is distributed in the peripheral region of the nucleus in an ordered, uniformly sized, focused, speckled pattern, which reflects the stable localization of chromosomal mitophagy [196]. However, in tumor cells, for instance, locoregional HNSCC (Head and Neck Squamous Cell Carcinoma), this orderly peripheral distribution pattern of CENP-A is severely disrupted and is scattered across the periphery of the nucleus in diffuse distribution. Interestingly, CENP-A patterns in radio-resistant HNSCC samples appear to be very heterogeneous, suggesting that different patterns of CENP-A localization may play a role in radiotherapy resistance [196].

#### 3.7.2. Prognostic Value of CENP-A

As CENP-A is ectopically expressed in a wide range of cancers, multiple studies have turned their attention to its prognostic value for patients’ overall survival rate, tumor stage and its potential to become a novel biomarker for early cancer detection [190,197,198].

Most of the reports support the idea that CENP-A levels can effectively predict patients’ pathological TMN stage and their recurrence-free survival due to the evidence provided by different research groups, confirming the high expression of CENP-A from their IHC and many other staining experiments [190,199]. However, as an upraised level of CENP-A is a common characteristic observed across numerous malignancies, it may not be a specifically effective indicator for specific types of cancer. Undeniably, CENP-A can serve as an excellent prognostic factor to predict patients’ overall survival rate and tumor stage, as the levels of CENP-A are positively correlated with the tumor progression status.

In order to achieve high sensitivity and specificity towards a particular type of cancer, the use of combinatorial indicators is highly recommended. Different cancers have their unique biomarkers, and as CENP-A somehow influences immune cell infiltration, infiltrated immune cells may also be utilized in the combinational use with CENP-A. This may hopefully screen for suspected and undetected tumor progression during early cancer stages [200,201,202,203,204]. However, in the later stage, a regular check-up using CENP-A alone as a biomarker can still monitor any deteriorating conditions in both treated and untreated patients, such as the high risk of distant metastasis and relapse of cancer [197,205,206]. Taken together, CENP-A is highly recommended to be interpreted as stage-dependent, and its specificities and sensitivities can be intensified when evaluated with other biomarkers.

## 4. Histone H4 Variants—Primate-Specific H4G

To date, only one novelly identified and emanated histone H4 variant named H4G. H4G is hominidae-specific and is encoded by gene *HIST1H4* located in the histone cluster 1 on human chromosome 6p22.1–22.2 [147,207]. H4G shares 85% identity with canonical H4 histone with a difference of 15 amino acid residues mainly near the N-terminal α1, α2 and α3 regions, and a shortened C-terminal by five residues [147,208,209].

H4G is mostly concentrated in the nucleolus at the molecular level. Subsequent research has revealed that the H4G amino acid residues, A85 and V89, are critical for the H4G-NPM1 (Nucleophosmin 1) interaction. This binding is therefore essential and permits H4G localization into the nucleolus [147,207]. Interestingly, follow-up experiments from Pang et al. revealed that H4G usually forms a temporary nucleosome-like structure and then rapidly disassembles [210]. This proves that the role of H4G in cells is probably to loosen up the chromatin structure and facilitate the rRNA (ribosomal RNA) transcription process and ribosome biogenesis. Regretfully, very little is known about this unique H4G variant—future research is required to determine other unknown roles it performs in normal cells.

### H4G—Role in Cancer

H4G is overexpressed mostly in breast cancer, followed by T-cell prolymphocytic leukemia, HuR-silenced thyroid carcinoma and endometrioid carcinoma [211,212,213]. Since H4G is newly identified, the corresponding functional roles of H4G in the later three cancers remain uncharacterized. In breast cancer, studies have found that H4G expression levels were positively correlated with cancer stage progression, as the knockdown of H4G in MCF7 cells led to a significant reduction in cell growth [207]. As previously mentioned, H4G plays a role in rRNA transcription and ribosome biogenesis. Studies have further illustrated that upon the depletion of H4G, cell proliferation, rRNA synthesis and protein synthesis dramatically decreased in breast cancer cells. Additionally, the functional role of H4G in breast cancer cell progression was confirmed by Long et al. using mouse xenograft models [207]. Altogether, these results strongly suggested that H4G loosens up the chromatin structure to facilitate rapid rRNA synthesis and continuously support the production of essential proteins required for cancer growth (Figure 5). Nevertheless, the mechanism behind the upregulated gene expression upon tumorigenesis, any alterations in downstream signaling pathways and any more unknown interacting proteins in vivo must be addressed and require more investigation.

## 5. Conclusions and Perspectives

Each canonical histone protein has its own group of variants. There are a total of 19 H2A variants (that have already been discussed in the previous review [20]), 14 H2B variants, 11 H3 variants and 1 H4 variant being identified in mammalian species to date (Table 1). Remarkably, all of these variants play unique pivotal roles in different kinds of intracellular mechanisms to safeguard genome stability and host immune system defense to protect hosts from viral/bacterial infections and disease development, including cancer progression. In recent decades, a slight increase in attention has been drawn to the identification of unrevealed variants in H2B, H3 and H4. However, the described details for these unprecedented variants can only be found in one paper published by Raman et al., which was limited to variants’ origins and structural differences with their corresponding canonical counterparts. Additionally, some novel variants, including H2B.F, H2B.M, H2B.Q and H2B.B, are solely described in a few sentences without any informative conclusions. Despite the lack of supporting evidence, we have tried our best and studied the most up-to-date references to discuss and list out all the reported roles of these mammalian variants here. We hope that these unknown topics can open up the possibility for future elucidation and provide us with insightful explanations for the presence of these variants in mammalian species.

In this review, we have organized the central core into three different parts: (1) Histone variants’ roles and localization in normal cell tissues in order to maintain genome stability; (2) Their functional roles in a broad range of malignancies and how they contribute towards cancer cell proliferation, invasiveness and metastasis; and (3) Their prognostic value and potential in becoming a biomarker for early cancer detection in the background of the use in clinical settings. Unexpectedly, H2B variants have all been identified as testes-specific variants, and the majority of them do not seem to participate in any of the cancer-related pathways. Only H2B.1 and H2B.2 have been outlined in short in a few papers published in 1989 and 1994, showing their dynamic expressions in Friend tumors. Alternatively, some H2B variants hold different roles besides spermatogenesis. For example, H2B.E and H2B.J control of viral infections, while H2B.A may contribute to diabetes development. While H2B variants may not take part in cancer progression, they may, however, influence the development of other human diseases, such as neurological diseases, due to their presence in the brain and neurons (H2B.E, H2B.1 and H2B.2). This suggests a new direction for future research and calls for more investigation to determine the validity of this hypothesis.

Most H3 variants are carcinogenesis-related, with CENP-A being the most well-studied in a broad variety of malignancies. CENP-A and other H3 variants’ functional roles in tumorigenesis are more or less the same to upregulate the cell-cycle related genes, enhance cell proliferation, escape apoptosis, increase cell migration and invasive ability and alter the tumor immune microenvironment. For the H4 variant, even though H4G was the only one uncovered and found to be involved in breast cancer progression, it is believed that with more research in the future, we will be able to unveil more information on the participation of H4G in other cancer types.

Finally, we believe that every variant possesses hidden potential to act as biomarkers or prognostic indicators in various human diseases and cancers. However, limited sensitivity and specificity for the detection of the variants may hinder their individual applications in clinical settings. Based on this, we strongly encourage the combinatorial use of well-defined biomarkers for a more precise and reliable diagnostic outcome. Moreover, research on histone variants should not only be restricted to cancers but should also focus on other incurable human diseases such as diabetes or neurological diseases. Therefore, the role of variants in human disease development should not be underestimated and thus deserves further investigation. As promising evidence is continuously offered in this field, it is anticipated that more novel insights and strategies for developing innovative treatments and identifying new variants as therapeutic targets will be provided in the future.

## Figures and Tables

**Figure 1 ijms-25-09699-f001:**
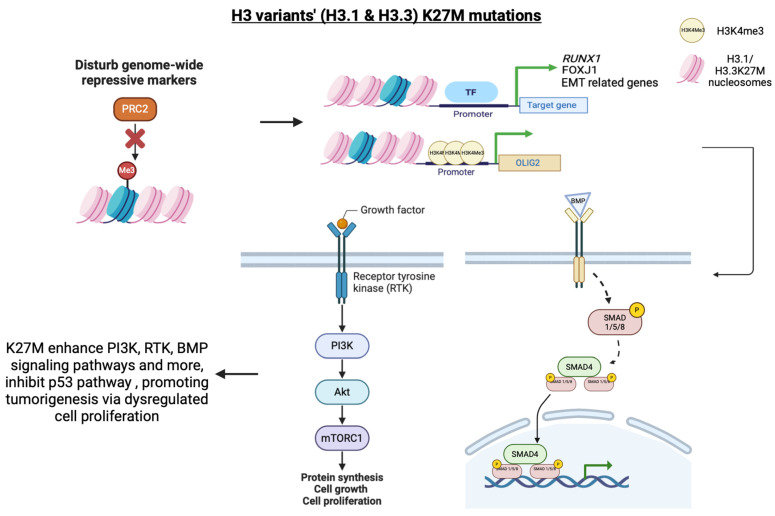
Diagram showing the consequences resulting from H3.1/H3.3-K27M mutation in tumor progression. Both the H3.1 and H3.3-K27M mutation was found to inhibit the genome-wide spread of H3K27me3 repressive marks on heterochromatin and stage-dependent silenced genes’ regions. The misregulated epigenetic modification patterns due to K27M mutation significantly prone normal cells to experience upregulation and aberrant activation and inhibition of multiple downstream signaling pathways (which cannot be listed and illustrated here) that result in promoting tumorigenesis, mainly in brain cancers. (Abbreviations: PRC2—Polycomb Repressive Complex 2, TF—Transcription factor, RUNX1—Runt-related Transcription Factor 1, FOXJ1—Forkhead Box Protein J 1, EMT—Epithelial–Mesenchymal Transition, OLIG2—Oligodendrocyte transcription factor, PI3K—Phosphatidylinositol-3-kinase, Akt—Protein kinase B, mTOR—mammalian Target of Rapamycin, BMP—Bone Morphogenetic Protein, SMAD—Mothers against decapentaplegic homolog 4).

**Figure 2 ijms-25-09699-f002:**
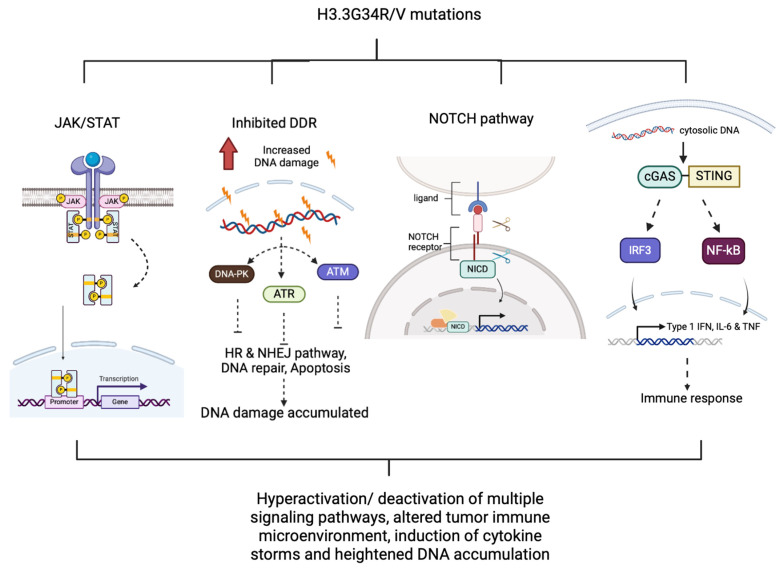
Diagram showing the altered downstream signaling pathways with H3.3G34R/V mutations. Upon H3.3G34R/V mutation, JAK/STAT, NOTCH and cGAS-STING signaling pathways are aberrantly hyperactivated with the deactivation of DNA damage response. This led to a significant accumulation of DNA mutations and enhanced gene transcription that are related to cell growth, migration, immune response and metastasis in cancer cells. (Abbreviations: HR—homologous repair, NHEJ—non-homologous end joining, DDR—DNA damage response, JAK/STAT—Janus Kinase/Signal Transducer and Activator of Transcription, ATR—Serine/threonine-protein kinase, ATM—Ataxia-telangiectasia mutated kinase, NOTCH—Neurogenic locus notch homolog protein 1, NICD—NOTCH intracellular domain, cGAS-STING—Cyclic GMP–AMP Synthase/Stimulator of Interferon Genes, IRF3—Interferon regulatory factor 3, NF-κB—Nuclear factor kappa-light-chain-enhancer of activated B cells).

**Figure 3 ijms-25-09699-f003:**
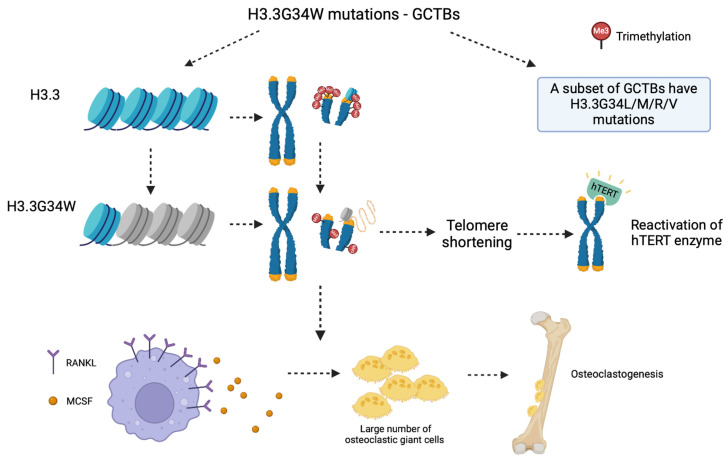
Schematic illustration depicting the intrinsic cell biology in GCTBs harboring H3.3G34W mutation. Telomere maintenance is worsened under the presence of the H3.3G34W variant through the hypomethylation of telomeric regions. The hTERT enzyme is also reactivated upon telomere shortening. On the other hand, H3.3G34W can increase the RANKL expressions on macrophages’ cell surface, which turns out to secrete more MCSF and recruit a larger amount of osteoclastic giant cells, leading to osteoclastogenesis. (Abbreviations: GCTBs—Giant Cell Tumor of Bone, hTERT—human Telomerase Reverse Transcriptase, RANKL—Receptor Activator of Nuclear Factor Kappa-B ligand, MCSF—Macrophage colony stimulation factor).

**Figure 4 ijms-25-09699-f004:**
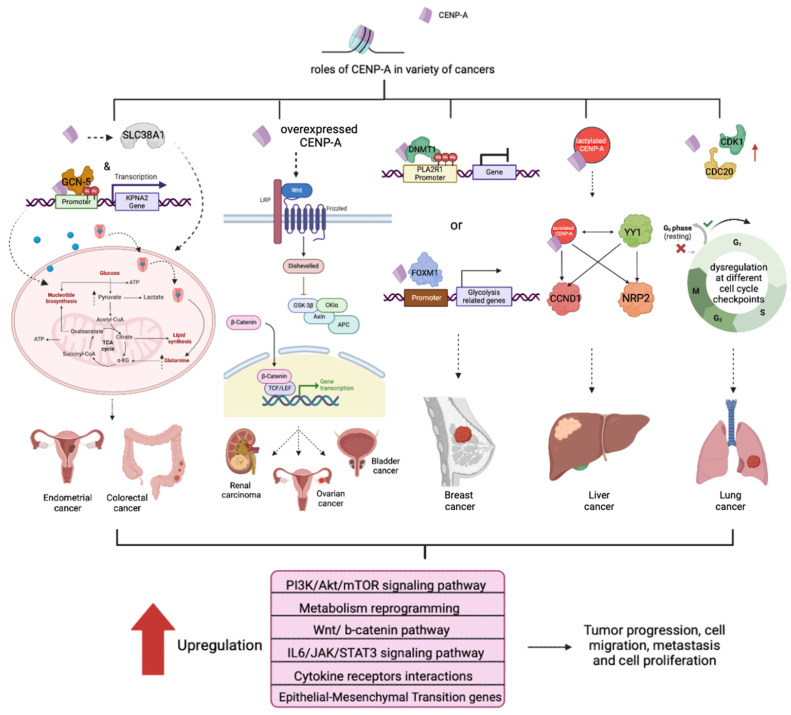
Schematic illustration showing how CENP-A plays a role in a broad variety of cancers. Overexpression of CENP-A is commonly observed, and it can work with CDC20 and CDK1 to help tumor cells escape from cell cycle checkpoints in lung adenocarcinoma. Modifications on CENP-A, such as lactylated, allow the cooperation with YY1 to produce CCND1 and NRP2 in HCC. In addition, CENP-A acts as a transcriptional regulator to enhance different gene expression in distinct cancers, including breast cancers, colorectal cancers and endometrial cancers that subsequently alter the metabolism reprogramming. Furthermore, CENP-A overexpression is found to correlate with hyperactivation of a wide range of downstream signaling pathways, such as Wnt/β-catenin in ccRCC, ovarian cancers and prostate cancers. (Abbreviations: SLC38A1—Solute Carrier Family 38 member 1, CCND1—cyclin D1, NRP2—Neuropilin 2, KPNA2—Karyopherin α2 subunit, DNMT1—DNA Methyltransferase 1, PLA2R1—Phospholipase A2 Receptor, FOXM1—Factor Forkhead Box M1, CDK1—Cyclin-dependent kinase 1, CDC20—Cell division cycle protein 20).

**Figure 5 ijms-25-09699-f005:**
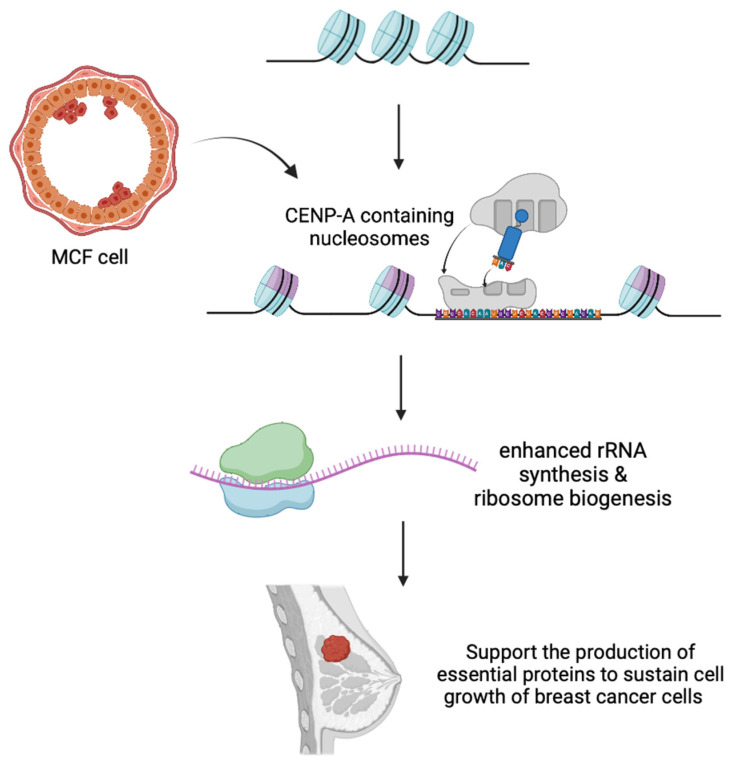
Schematic illustration depicting the role of H4G in breast cancer progression. H4G is demonstrated to enhance chromatin accessibility and facilitate rRNA synthesis and ribosome biogenesis. This allows the continued production of essential proteins to support breast tumor growth. (Abbreviations: MCF—Michigan Cancer Foundation, rRNA—ribosomal RNA).

**Table 1 ijms-25-09699-t001:** All the documented mammalian histone variants from canonical histone H2B, H3 and H4 families to date are listed in the table. Alternative names are written in the brackets. There are a total of 14 H2B variants, 11 H3 variants and 1 H4 variant.

Core Histone (Canonical Histone)	Histone Variants
H2B	H2B.A
	H2B.B
	H2B.E
	H2B.F
	H2B.J
	H2B.K
	H2B.L (H2BP4/sub H2B)
	H2B.M
	H2B.N
	H2B.O
	H2B.Q
	H2B.W
	H2B.1 (TSH2B/hTSH2B/TH2B)
	H2B.2
H3	H3.1
	H3.2
	H3.3
	H3.4 (TSH3.4/H3.1t/H3T)
	H3.5
	H3.6
	H3.7
	H3.8
	H3.Y.1 (H3.Y)
	H3.Y.2 (H3.X)
	CENP-A (CENH3)
H4	H4G (H4.7)

## Data Availability

Not applicable.

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
