# Peer review of "Roles of Histone H2B, H3 and H4 Variants in Cancer Development and Prognosis"

_ijms, 2024, doi:10.3390/ijms25179699_

Round 1

Reviewer 1 Report

Comments and Suggestions for Authors

This is an interesting review on the roles of histone H2B, H3 and H4 variants may have on cancer. In general, the manuscript is clear and well organized. However, some points should be addressed to improve the quality of the paper. Some suggestions are the following:

Line 49: I think it would be useful to explain the meaning of the word “paralogs”

Lines 52-54: This sentence is not clear to me. Please explain what “proteomic features” are when referring to a protein molecule

Lines 90-92: From this sentence, it seems that the H2B variants have nothing to do with cancer development and prognosis. Instead, they have a role in spermatogenesis and early fertilization. This is in contrast with the title of the review.

Line 98: Why start the description of histone H2B variants from H2B.E? As reported in the section 2.1., this variant has nothing to do with cancer! Similarly, both H2B.A and H2B.W variants have nothing to do with cancer. Why not start with the H2B.1 variant, which seems to have a correlation with pediatric brain tumor?

Line 179-183: The meaning of these sentences is not clear. As above, the rare H2B variants have little to do with cancer.

The rest of the manuscript on both H3 and H4 variants is clear and well describe also by the different figures.

Comments on the Quality of English Language

Moderate editing of English language required. A few sentences are not clear.

Author Response

Line 49: I think it would be useful to explain the meaning of the word “paralogs”

Answer: Thank you. I have taken a lot of literatures as references, and it seems that “paralogs” is a common and understandable term used in the epigenetic field.

Lines 52-54: This sentence is not clear to me. Please explain what “proteomic features” are when referring to a protein molecule

Answer: Thank you for your suggestions, wordings have been changed accordingly.

Lines 90-92: From this sentence, it seems that the H2B variants have nothing to do with cancer development and prognosis. Instead, they have a role in spermatogenesis and early fertilization. This is in contrast with the title of the review.

Answer: Thanks for your comment. However, I would like to make a complete story for all the histone families, even though H2B variants didn’t seems to have much correlation with cancer development. It is still possible that people used to its function in spermatogenesis and may choose to ignore their expression levels in other cancer types/ other organs (without deeper examination), which hinder H2B variants’ possible participation in different cancers. Therefore, I would like to point it out and calls for more investigation in H2B variants by more experts.

Line 98: Why start the description of histone H2B variants from H2B.E? As reported in the section 2.1., this variant has nothing to do with cancer! Similarly, both H2B.A and H2B.W variants have nothing to do with cancer. Why not start with the H2B.1 variant, which seems to have a correlation with pediatric brain tumor?

Answer: Thank you for your comments. I arranged in this order is depended on 1) the number of published papers I can find in PubMed, 2) their functional role described in different scenario and 3) the detail information listed within each literature with references to each individual variant. H2B.E has a comparatively more comprehensive description than the other variants, that’s why I started with H2B.E first.

Line 179-183: The meaning of these sentences is not clear. As above, the rare H2B variants have little to do with cancer.

Answer: Thank you for pointing this out. These rare H2B variants are newly identified without much elucidation, we believed that they may have some uncovered relation with cancer progression, but this remains elusive in the future.

Response: Thank you so much for all the valuable and positive advice for the manuscript. Thank you for pointing out your confusion, as I would like to complete this story as a whole without ignoring one of the histone classes, that’s why H2B variants will still be present in the manuscript. All the corrected parts are highlighted in yellow. Thank you once again for your time. 

Reviewer 2 Report

Comments and Suggestions for Authors

In the present manuscript, the authors present a very comprehensive review on the histone variants, including the paralogs of all core histones. Including most up-to-date references, they discuss their biological role, but also their potential as diagnostic markers, and the efforts to address these complex molecules as therapeutic targets. The review is very detailed and summarizes a large amount of information, however it is written in contemporary lively language, which makes it interesting to read. The Figures are well designed and a good support to text. My only remarks would be that few sentences are not completely conclusive (please see the  remarks below). Legends to the Figures could be improved by including the explanations of abbreviations used therein.

Certain references (e.g. 16, 107, 146, 149, 151) are not in the standard format.

Certain references repeat (26/41, Siuti et al.)

Please find below a list of remarks which I hope will be helpful.

Line 185: at a low enrichment

Line 187: its clear absence

Line 258: the majority of published studies

Line 263: between the two cysteines

Line 269: deposition is regulated

Lines 316-318: this sentence is very difficult to understand

Line 341: thermosensation is the more usual expression

Lines 350-351: 358th amino acid Ile to Leu (A to C) mutation

Line 380: significantly alter normal cells

Line 382: which results in promoting

Line 433: redox sensor

Line 551, Figure 2: Abbreviations such as HR and NHEJ should be explained in the Figure Legend.

Line 637:” Where GPR87 has shown its potential in…” this sentence is incomplete, please correct.

Lines 675-681: very important observations, unfortunately the citations are missing

Line 696-697:  and in line with this, it has a transparent and no spermatozoa was found in the epididymides” – this sentence is not complete

Line 732: “Whether will H3.5 taking over the role of H3.3 and contribute to…” – the sentence is not complete

Lines 775-777: “for future elucidation, providing fresh perspectives on the pathogenesis or aetiology of certain diseases and disorders.“: this paragraph is too generic: maybe reword to: Future elucidation of their importance, as well as of the consequences of their mutational or post-translational modifications, will provide fresh perspectives… or similar.

Line 847: CENPA should be CENP-A, please correct the hyphenation throughout the article

Figure 4: organ sketches require labelling, and all abbreviations (including the names of the genes) should be explained in the Figure legend. Regarding the excellent incorporation of the Figure to support the text, the pertaining references could be indicated in the Figure and indexed in the Figure legend.

Lines 949-951: “Undeniably, CENP-A can serve as an excellent prognostic factor to predict patients’ overall survival rate and tumor stage, as the levels of CENP-A are positively correlated with the tumor progression status. “ – could be supported by additional references

Line 961-962: “…recommended to be stage–dependent and coupled with other biomarkers to intensify its specificities and sensitivities”: the scientific language of this sentence is not at the same level with the rest of the article, and this statements should be less general: certainly, you mean that the positive evidence of CENP-A is recommended to be interpreted as stage-dependent, and further, its significance can be enhanced when evaluated together with other biomarkers. This passage is of high translational importance and please render its significance as straight-forward as possible.

Line 1002: „…in the previous review“ – please insert the reference

Line 1013:” Despite the lack of supporting evidence, we have tried our best…” – maybe reword to: we have studied the most up-to-date references to discuss…”

Line 1028: “may however, be playing around in the” – they may influence;  playing around is too colloquial

Line 1038:” is the only one uncovered and involved in breast cancer progression”- please reword to: is the only one uncovered and to that, found to be involved in breast cancer progression, or similar

Line 1042:” limited sensitivity and specificity of the variants” – the statement is too general; do you refer to limited sensitivity and specificity of the detection?

Line 1049: deserves further investigation

Lines 1050-1051: this is a challenging goal which deserves to be split into two sentences, one on the identification of novel variants as therapeutic targets and a second describing development of therapies involving histone targets

Author Response

Certain references (e.g. 16, 107, 146, 149, 151) are not in the standard format.

Certain references repeat (26/41, Siuti et al.)

Response: Thank you so much for your reminder. The repeated references and format have been amended accordingly.

Please find below a list of remarks which I hope will be helpful.

Line 185: at a low enrichment

Line 187: its clear absence

Line 258: the majority of published studies

Line 263: between the two cysteines

Line 269: deposition is regulated (I was referring to both H3.1 and H3.2 deposition, so I think it will be better to use “are”)

Lines 316-318: this sentence is very difficult to understand (Thanks for the comment. I have rephrased the sentence accordingly)

Line 341: thermosensation is the more usual expression

Lines 350-351: 358th amino acid Ile to Leu (A to C) mutation

Line 380: significantly alter normal cells

Line 382: which results in promoting

Line 433: redox sensor

Line 551, Figure 2: Abbreviations such as HR and NHEJ should be explained in the Figure Legend. (Thanks for your reminder)

Line 637:” Where GPR87 has shown its potential in…” this sentence is incomplete, please correct. (Thank you, I have changed the starting word)

Lines 675-681: very important observations, unfortunately the citations are missing (Thank you, I added back the citation)

Line 696-697:  and in line with this, it has a transparent and no spermatozoa was found in the epididymides” – this sentence is not complete

Line 732: “Whether will H3.5 taking over the role of H3.3 and contribute to…” – the sentence is not complete (This sentence is completed, and I have mentioned that this remains elusive)

Lines 775-777: “for future elucidation, providing fresh perspectives on the pathogenesis or aetiology of certain diseases and disorders.“: this paragraph is too generic: maybe reword to: Future elucidation of their importance, as well as of the consequences of their mutational or post-translational modifications, will provide fresh perspectives… or similar. (Thank you for the suggestion, I will take this as references)

Line 847: CENPA should be CENP-A, please correct the hyphenation throughout the article

Figure 4: organ sketches require labelling, and all abbreviations (including the names of the genes) should be explained in the Figure legend. Regarding the excellent incorporation of the Figure to support the text, the pertaining references could be indicated in the Figure and indexed in the Figure legend.

Lines 949-951: “Undeniably, CENP-A can serve as an excellent prognostic factor to predict patients’ overall survival rate and tumor stage, as the levels of CENP-A are positively correlated with the tumor progression status. “ – could be supported by additional references 

Line 961-962: “…recommended to be stage–dependent and coupled with other biomarkers to intensify its specificities and sensitivities”: the scientific language of this sentence is not at the same level with the rest of the article, and this statements should be less general: certainly, you mean that the positive evidence of CENP-A is recommended to be interpreted as stage-dependent, and further, its significance can be enhanced when evaluated together with other biomarkers. This passage is of high translational importance and please render its significance as straight-forward as possible. (Thank you for such a valuable comment. The sentence has rephrased accordingly.)

Line 1002: „…in the previous review“ – please insert the reference

Line 1013:” Despite the lack of supporting evidence, we have tried our best…” – maybe reword to: we have studied the most up-to-date references to discuss…”

Line 1028: “may however, be playing around in the” – they may influence;  playing around is too colloquial (Thank you for the suggestion. Wordings have changed in the sentence.)

Line 1038:” is the only one uncovered and involved in breast cancer progression”- please reword to: is the only one uncovered and to that, found to be involved in breast cancer progression, or similar

Line 1042:” limited sensitivity and specificity of the variants” – the statement is too general; do you refer to limited sensitivity and specificity of the detection?

Line 1049: deserves further investigation

Lines 1050-1051: this is a challenging goal which deserves to be split into two sentences, one on the identification of novel variants as therapeutic targets and a second describing development of therapies involving histone targets

Response: Thank you so much for all the valuable and useful comments and sentence-rephrase suggestion. I have factored all the comments into consideration and have amended in my manuscript accordingly. Thank you for all the sharp observation with regarding to my references lists (formats and repeated reference), these mistakes were also corrected. Thank you for your reminder in asking to add the abbreviations in the figure legend to enhance the quality of this manuscript. All the corrected parts are highlighted in yellow
